# Changes of Physicochemical, Bioactive Compounds and Antioxidant Capacity during the Brewing Process of Zhenjiang Aromatic Vinegar

**DOI:** 10.3390/molecules24213935

**Published:** 2019-10-31

**Authors:** Wenhui Duan, Ting Xia, Bo Zhang, Shaopeng Li, Chenwei Zhang, Chaoya Zhao, Jia Song, Min Wang

**Affiliations:** 1State Key Laboratory of Food Nutrition and Safety, Ministry of Education, Tianjin Engineering Research Center of Microbial Metabolism and Fermentation Process Control, College of Biotechnology, Tianjin University of Science and Technology, Tianjin 300457, China; duanwhyy@163.com (W.D.); xiatingsyu@foxmail.com (T.X.); bozhango@163.com (B.Z.); li17627820568@163.com (S.L.); zcw0414@163.com (C.Z.); zcysuccess@foxmail.com (C.Z.); tjsongjia@tust.edu.cn (J.S.); 2Key Laboratory of Industrial Fermentation Microbiology, Ministry of Education, Tianjin Engineering Research Center of Microbial Metabolism and Fermentation Process Control, College of Biotechnology, Tianjin University of Science and Technology, Tianjin 300457, China

**Keywords:** Zhenjiang aromatic vinegar, brewing process, antioxidant activity, bioactive compounds, phenolics, functional food

## Abstract

Zhenjiang aromatic vinegar (ZAV) is a kind of traditional fermented food worldwide. In this study, the changes of physicochemical properties, total phenolic content (TPC), total flavonoid content (TFC), and total antioxidant activity (TAA) were evaluated during the brewing process of ZAV. In addition, the correlation between phenolic compound contents and antioxidant activities was investigated during the aging process (AP) of ZAV. The results showed that total acids, non-volatile acids, and amino nitrogen increased gradually during the brewing process. Reducing sugar decreased sharply at the early fermentation stage and then increased during the AP. Meanwhile, TPC, TFC, and TAA kept a very low level at the stage of alcohol fermentation (AF), and increased to the highest level at the sixth year of the AP. TAA has a high positive correlation with TPC and TFC during the brewing process of ZAV. In addition, the contents of *p*-hydroxybenzoic acid, vanillic acid, and catechin were higher than other phenolic compounds and reached the highest level at the sixth year of the AP, and were the main composition of phenolic compounds during the AP. Moreover, gallic acid, ferulic acid, and sinapic acid had the higher contribution at the early stage of the AP, and then declined to a lower level. Catechin, vanillic acid, and syringic acid had a higher contribution during the AP. These findings would be helpful in controlling the quality of vinegar and improving its functional properties.

## 1. Introduction

Vinegar has been widely used as an acidic condiment all over the world and is a kind of fermented and functional food [1]. Vinegar can be classified into fruit vinegar and grain vinegar according to different raw materials [2]. Most fruit vinegars are produced by liquid-state fermentation and are mainly distributed in Europe and Africa. These vinegars are made from grapes, apples, tomatoes, persimmons, and pineapple, which are famous as balsamic vinegar, Sherry vinegar, and apple vinegar [3,4,5]. Most grain vinegars are mainly brewed by solid-state fermentation in Asia, such as Kurozu vinegar, Shanxi aged vinegar, Zhenjiang aromatic vinegar (ZAV), and Baoning vinegar. Raw materials of these vinegars are made of sorghum, wheat bran, beans, rice, and rice hulls [6,7]. ZAV, one of the famous Chinese vinegars, is made of grains such as sticky rice, barley, wheat bran, and rice hulls [8]. ZAV is produced by a solid-state fermentation method, including saccharification and alcoholic fermentation (AF), acetic acid fermentation (AAF), leaching, decoction, and aging [9]. During the AF, the raw materials are crushed, cooked, saccharificated, and fermented. AAF of ZAV is a typical process of multispecies solid-state fermentation. Vinegar *Pei*, a special term in AAF of Chinese vinegars, refers to a mixture of alcohol mash, raw materials, and complex bacterial communities [10]. During the AAF, the Pei at the seventh day is inoculated at the surface of fresh *Pei.* Meanwhile, the method of layered fermentation is applied using an automatic turning machine, which turns the *Pei* down a layer every day and to the bottom of the fermented pool at the seventh day of AAF. This method is beneficial in expanding the microorganisms and making full use of raw materials [11]. The contents of the products are increased due to the evaporation of the water during the aging process (AP). Meanwhile, many macromolecules are produced by biochemical reactions during this process, which improve the flavor and functional components, such as melanoidins and tetramethylpyrazine [12,13].

There are many physicochemical and bioactive compounds in vinegar, which are important parameters to monitor and control vinegar quality during the brewing process [14,15]. Total acids and pH are the key indicators to assess vinegar quality, which should be more than 6 °T and from 3.6 to 3.9 in Shanxi aged vinegar, respectively (GB/T19777-2013). In addition, a variety of bioactive compounds, including polyphenols and flavonoids, are changed during the brewing process [16]. The phenolic compounds in vinegar are mainly derived from raw materials and secondary metabolites produced by microorganisms during the fermentation process [17]. Several studies have reported that vinegar is rich in antioxidant ingredients, including phenols and flavonoids, which can terminate the free-radical chain reaction and chelate with metal ions [17,18]. Ozturk et al. [19] reported that the total phenolic content (TPC) and total flavonoid content (TFC) in traditional Turkish vinegar were 40.44 ± 2.58~2228.79 ± 83.26 mg/L and 10.89 ± 0.59~349.05 ± 2.80 mg/L, respectively. Also, 1,1-diphenyl-2- picrylhydrazyl (DPPH) scavenging activities varied from 4.93 ± 0.12% to 89.91 ± 0.65%. It has been reported that the antioxidant capacity of foods is highly correlated with their TPC and TFC [20,21]. Porgal et al. [22] reported that TPC and total antioxidant activity (TAA) in red wines were 1836.5 ± 40.5~3466.9 ± 54.4 mg/L and 284.5 ± 11.3~774.7 ± 14.4 mg GAE/L, respectively. There was a very high degree of correlation between TPC and TAA. Xia et al. [23] found that TPC, TFC, and antioxidant activity of Shanxi aged vinegar were increased with the aging time. TPC (3.732 ± 0.329 mg GAE/mL), TFC (3.425 ± 0.510 mg RE/mL), and antioxidant activity in 2,2’-azino-bis(3-ethylbenzthi azoline-6-sulfonic acid) (ABTS) assay (6.10 ± 0.62 mmol Trolox equivalent antioxidant capacity (TEAC)/L~29.94 ± 0.58 mmol of TEAC/L) in vinegar aged for 8 years were the highest. TPC and TFC showed a high correlation with TAA (r = 0.929, r = 0.879, *P* < 0.01, respectively). However, the changes during the brewing process of ZAV are rarely explored.

Phenolic compounds in vinegar scavenge free radicals and exhibit antioxidant capacity, which have benefits for the human body [24,25,26]. Plessi et al. [27] reported that eight phenolic compounds were detected in traditional balsamic vinegar by gas chromatography-mass spectrometer (GC-MS). Protocatechuic acid, gallic acid, and *p*-coumaric acid were the main phenolic acids in traditional balsamic vinegar. Bakir et al. [28] reported that grape vinegar contained two phenolic compounds (gallic acid and *p*-hydroxybenzoic acid), and apple vinegar contained six phenolic compounds (gallic acid, *p*-hydroxybenzoic acid, catechin, caffeic acid, and *p*-coumaric acid). The antioxidant activities of grape vinegar and apple vinegar ranged from 418 ± 49 mg TEAC/100 mL to 2561 ± 260 mg TEAC/100 mL by ABTS assay. The result indicated that the phenolic compounds in vinegar had strong antioxidant capacities. These studies have reported the kinds and contents of phenolic compounds and TAA in vinegar [29,30]. However, the changes of phenolic compounds and antioxidant characteristics during the brewing process of ZAV are rarely explored.

In this study, we investigated the variation of physicochemical parameters in ZAV during the brewing process. Meanwhile, we characterized the changes of TPC, TFC, and TAA during the brewing process. Furthermore, the kinds and contents of phenolic compounds and their correlation with antioxidant activity were examined during the AP. This study would be helpful to monitor and control antioxidant properties during the vinegar production process.

## 2. Results and Discussion

### 2.1. Changes in Physicochemical Parameters during the Brewing Process of ZAV

Total acids are an important indicator for assessing the quality of vinegar. Reducing sugars and amino nitrogen play an important role in the evaluation of food quality [31,32]. As shown in Figure 1, the changes of general components in ZAV were analyzed during the brewing process. At the stage of AF, the contents of total acids, non-volatile acids, and amino nitrogen were at a low level, and showed a gradual increase. In addition, the content of reducing sugar decreased rapidly from 7.68 ± 0.02 g/100 mL to 2.64 ± 0.01 g/100 mL, and then declined slowly. The pH values reduced gradually from 4.72 to 4.52 at this stage. During AAF, the contents of total acids, non-volatile acids, and amino nitrogen showed a gradual increase, and the contents were about 3.27, 2.01, and 2.10 times, respectively, at the 20th day, as many as those at the beginning of this stage. The pH values decreased gradually due to the increasing organic acids, which ranged from 3.56 ± 0.00 to 4.42 ± 0.01. The content of reducing sugar gradually decreased with the fermentation progress and increased slightly at the end of AAF. During the AP, the contents of total acids, non-volatile acids, amino nitrogen, and reducing sugar increased obviously at the first two days, and then elevated slowly. The pH values were stable, which were maintained between 3.38 ± 0.00 and 3.470 ± 0.00.

### 2.2. Alteration of TPC and TFC during the Brewing Process of ZAV

Polyphenols and flavonoids are the main bioactive compounds in vinegar, which primarily came from raw materials [33]. In this study, the TPC and TFC of ZAV samples during the production process were determined respectively. As shown in Figure 2, the results showed that TPC and TFC were at a very low level during the stage of AF, indicating that the raw materials had not been fully utilized at this stage. Then, at the stage of AAF, the TPC and TFC showed a gradual increase and a slight decline at the 13th day, and then increased stably and reached the highest level on the 19th day of fermentation stage. These data suggest that polyphenols and flavonoids began to accumulate during the process of AAF. Finally, during the AP, the TPC and TFC ranged from 2.072 ± 0.008 mg GAE/mL to 4.066 ± 0.112 mg RE/mL, 1.206 ± 0.052 mg GAE/mL to 2.999 ± 0.066 mg RE/mL, respectively. The results showed that TPC and TFC increased obviously with the aging time. These contents were at the highest level at the sixth year of the AP, and then a slight decrease at the seventh year of the AP. Xie et al. [21] reported that the TPC and TFC in Shanxi aged vinegar were 0.96~5.80 mg GAE/mL and 0.33~4.50 mg RE/mL, respectively. The TPC and TFC were obviously increased with the aging time, which reached the highest level at the six-year-old aging time and declined slightly at the seven-year-old aging time. It has been reported that the increased phenolic content in traditional balsamic vinegar and sherry vinegar is due to the extraction of some phenolic compounds from the wood and evaporation of the water during the aging process [20,34]. In addition, flavonol glycosides were hydrolyzed into flavonol aglycones during the AP, which can increase the flavonoid content with the aging time [35]. Briefly, the increase of TPC and TFC is likely associated with the reduction of water content and hydrolyzation of flavonol glycosides during the AP.

### 2.3. TAA and its Correlation with TPC and TFC during the Brewing Process of ZAV

The bioactive compounds in vinegar have antioxidant activities [36,37]. Studies have reported that TPC and TFC were highly correlated with TAA of vinegars [38,39]. Next, the antioxidant activities of ZAV samples were measured by DPPH, ABTS, and ferric reducing antioxidant power (FRAP) methods, as shown in Figure 3. The results showed that the antioxidant activities were very low at the stage of AF, and then fluctuated during AAF. During the AP, the antioxidant activities increased markedly during six years of aging time, and then decreased slightly at the seventh year of the AP. Collectively, the trend of TAA during the AP was similar to that of TPC and TFC. Bertelli et al. [40] reported that the antioxidant activities of traditional balsamic vinegar from Modena were 33.04 ± 21.8 μM TEs/mL (>25 years of aging) and 33.52 ± 19.3 μM TEs/mL (>12 and <25 years of aging), and that of balsamic vinegar from Modena were 16.88 ± 13.6 μM TEs/mL by ABTS assay. These results found that TAA in vinegar was increased with aging time, and then decreased slightly with higher aging time, which were in line with our data. During the AP, the low pH value and the decrease of the water content may promote the formation of the oligomers and polymers, which subsequently evolve to more polymerized compounds and finally precipitate [41]. Several studies have reported that phenolic compounds can interact with macromolecules, such as melanin, proteins, and polysaccharides, to form precipitates in vinegars during the AP [42,43]. Therefore, the decrease of TPC and TFC may be due to the increased precipitation during the AP, which subsequently declined the TAA at this stage.

The relationship between TPC, TFC, and TAA during the brewing process was analyzed by DPPH, FRAP, and ABTS in Table 1. The results showed that the correlations between TPC and TAA were 0.954 (*P* < 0.01), 0.987 (*P* < 0.01), and 0.952 (*P* < 0.01), respectively. Meanwhile, the correlations between the TFC and TAA were 0.951 (*P* < 0.01), 0.989 (*P* < 0.01), and 0.951 (*P* < 0.01), respectively. Verzelloni et al. [20] found that the antioxidant capacities of traditional balsamic vinegar and balsamic vinegar were highly correlated with their TPC and TFC. Alonso et al. [44] reported that the TPC of the Spanish sherry vinegar was 200~1000 mg/L, which showed a high correlation with the antioxidant capacity (r = 0.9201). Our results showed that TPC and TFC had a highly positive correlation with TAA, which was in accordance with previous studies. Taken together, the antioxidant capacity of ZAV was mainly based on the polyphenols and flavonoids, which increased obviously during the AP.

### 2.4. Variation of Phenolic Compound Contents during the AP of ZAV

Phenolic compounds in ZAV during the AP were separated and quantified by the high performance liquid chromatography (HPLC) method. As shown in Figure 4, the total amount of phenolic compounds increased with the aging time, and then decreased at the seventh year of aging. The contents of *p*-hydroxybenzoic acid, vanillic acid, and catechin were higher than other phenolic compounds. The contents of *p*-hydroxybenzoic acid, vanillic acid, and catechin in ZAV at the beginning of the AP were 3.440 ± 0.174 mg/L, 9.213 ± 0.962 mg/L, and 9.286 ± 0.632 mg/L, respectively, and then increased gradually. The contents of the three phenolic compounds reached the highest level (27.061 ± 2.689, 148.378 ± 11.982, and 57.453 ± 3.736 mg/L, respectively) at the sixth year of the AP, and then decreased. As shown in Appendix A, the browning index (A420 nm) of high molecular weight melanoidins ranged from 0.26 ± 0.01 to 0.83 ± 0.03, which reached the highest level at the sixth year of aging and decreased slightly at the seventh year of aging. These results were in accordance with our previous study [23], which may be associated with the reduction of water content during the AP. During the AP, the compositions of vinegar were concentrated and had undergone a series of chemical reactions after temperature changes in winter and summer. The Maillard reaction is one of the main reactions in the AP [45]. The increase of the browning index indicates that the Maillard reaction continues and the synthesis of polymeric compounds increases during the AP of six years. The Maillard reaction is a non-enzymatic browning reaction, and phenolic compounds were also involved in the formation of melanoidins [46]. Therefore, the content of phenolic compounds in ZAV aged for seven years was lower than that in ZAV aged for five years, while the seven-year aged ZAV had similar TAA to the five-year aged ZAV.

### 2.5. Antioxidant Contribution of Phenolic Compounds during the AP of ZAV

In order to further explore the contribution of phenolic compounds to the TAA during the AP, antioxidant activities of phenolic compounds were detected by DPPH, ABTS, and FRAP assays. As shown in Figure 5A, the contribution of phenolic compounds to the TAA mainly came from catechin, rutin, gallic acid, syringic acid, sinapic acid, and ferulic acid at the beginning of the AP. Among these phenolic compounds, gallic acid, ferulic acid, and sinapic acid showed the higher contribution during the early years of aging, and then declined to a lower level. Meanwhile, the contribution rates of catechin, vanillic acid, and syringic acid increased to the highest level at the fifth or sixth year of the AP, and then decreased slightly. Finally, phenolic compounds including catechin, vanillic acid, and syringic acid contributed mainly to the TAA at the seventh year of the AP (DPPH assay). In ABTS and FRAP assays, the trends of the phenolic compound contribution to the TAA during the AP were similar to those in the DPPH assay, as shown in Figure 5B,C. In addition, the whole TAA of mixed solution was higher at the early stage of the AP, and then declined after being aged two years. This may be due to some phenolic compounds being involved in the formation of melanoidins, which led to the decrease of contribution to phenolic compounds during the AP.

## 3. Materials and Methods

### 3.1. Samples and Reagents

ZAV samples during the brewing process were obtained from Jiangsu Hengshun Vinegar Industry Co., Ltd. (Jiangsu, China), as shown in Table 2. There were four samples per time point, and each sample with three replicates to measure standard errors. HPLC standards of phenolic compounds were purchased from Sigma Aldrich (Deisenhofen, Germany). Folin–Ciocalteu and DPPH reagents were purchased from Sinopharm Chemical Reagent Co., Ltd. (Shanghai, China). Antioxidant capacity assay kits with ABTS and FRAP were obtained from the Beyotime Institute of Biotechnology (Shanghai, China).

### 3.2. Detection of the Physicochemical Parameters

The pH of ZAV samples was determined by a pH meter (Metrohm, Herisau, Switzerland). Total acids were examined according to the GB/T 12456-2008. The content of amino nitrogen in vinegar samples was detected by formaldehyde titration (GB/T 5009.235-2016). The reducing sugar content in ZAV was calculated by Fehling’s method. The contents of non-volatile organic acids and soluble solids were detected according to the methods of the National Standard (GB/T18623-2011; GB/T18187-2000).

### 3.3. Measurement of TPC and TFC

The TPC of ZAV samples was determined by the modified Folin–Ciocalteu method [47]. Briefly, 0.2 mL of diluted sample was added to 0.8 mL of Folin–Ciocalteu reagent. After incubating for 3–5 min at room temperature, 1.5 mL of 10% Na_2_CO_3_ (*w*/*v*) solution was added and diluted to 10 mL with deionized water. After incubating for 120 min in the dark, the mixture was measured at 765 nm. Gallic acid was regarded as a standard. The results were expressed as mg GAE/mL.

The TFC of ZAV samples was determined by colorimetric assay. ZAV samples were neutralized with 2% NaOH solution (*w*/*v*) and diluted with deionized water. Then, 2 mL of diluted samples were mixed with 8 mL of deionized water and 1 mL of 5% NaNO_2_ solution (*w*/*v*). One milliliter of 5% Al(NO_3_)_3_ solution (*w*/*v*) was added and stood for 6 min. Finally, 4 mL of 20% NaOH solution (*w*/*v*) was added and diluted to 25 mL with distilled water. After incubation for 15 min in the dark, the absorbance was detected at 510 nm. Rutin was used as a standard. The data were expressed as mg RE/mL.

### 3.4. Determination of TAA

The antioxidant activities of ZAV samples were evaluated by three assays. In the DPPH assay, 20 μL of diluted samples and 180 μL of DPPH working solution were sufficiently mixed. The absorbance was measured at 517 nm with a microplate reader after incubating for 30 min at room temperature in the dark. A calibration curve was prepared with different concentrations of Trolox solutions. The results were performed as mmol TEAC/L of vinegar.

In the ABTS assay, an aqueous solution of ABTS and oxidant were mixed, in which ABTS radical cation (ABTS+) was generated. These reagents were kept to react completely in the dark for 12 h and then diluted with ethanol to an absorbance of 0.70 ± 0.05 at 414 nm. Then, 10 μL diluted samples were mixed with 200 μL ABTS working solution. The absorbance was measured at 414 nm in a microplate reader after 2–6 min in the dark. Trolox was used as a standard compound. The TAC was expressed as mmol TEAC/L of vinegar.

In the FRAP assay, preparation of a working solution was mixed with tripyridyltriazine (TPTZ), buffer solution, and the diluted solution at a ratio of 1:1:10 (*v*/*v*/*v*). Then, 5 μL of diluted sample was added to 180 μL of FRAP working solution. The absorbance was recorded at 593 nm with the microplate reader after incubating at 37 °C for 5 min. Trolox was used as a reference compound. The results were calculated as reported as mmol TEAC/L of vinegar.

### 3.5. Identification of Phenolic Compositions

Phenolic compositions of ZAV were determined by the HPLC method. Each sample was adjusted to pH 2.0 with 1 mol/L HCl solution and then ultrasonically extracted with ethyl acetate three times. Subsequently, the extracts evaporated to dry with rotary evaporation (Heidolph Digital, Schwabach, Germany), and then the residue was dissolved in 600 μL of 50% methanol (*v*/*v*). The solution was filtered through 0.45 µm Millipore membrane (Billerica, MA, USA) and injected into the HPLC system (Agilent Technologies Inc., California, CA, USA) with the Phenyl chromatography column (250 × 4.6 mm i.d., 5 μm, Nano-Micro, Suzhou, China). The following gradient system including water/acetic acid (98:2, *v*/*v*, solvent A) and water/acetonitrile/acetic acid (73:25:2, *v*/*v*/*v*, solvent B) was used as follows: 0 min, 5% B; 40 min, 30% B; 45 min, 20% B; 50 min, 100% B; 51 min 5% B, and then held for 5 min. The flow rate was 1.00 mL/min and the column temperature was 40 °C. The UV-vis detection wavelength was 278 nm. The major phenolic compounds were identified through comparison with the data of phenolic standards. The samples were tested 6 times by the HPLC methods.

### 3.6. Determination of High Molecular Weight Melanoidins

Vinegar samples were diluted and filtered with filter papers 40 (Whatman, Maidstone, UK). Each filtered sample was ultra-filtrated with Amicon Ultra-4 regenerated cellulose 10 kDa (Millipore, MA, USA) and centrifuged at 7500× *g* for 20 min at 4 °C. High molecular weight materials were isolated from each filtered sample. The amount of high molecular weight melanoidins was determined as the browning index. The absorbance was detected at 420 nm.

### 3.7. Analysis of Antioxidant Contribution of Phenolic Compounds in Mixed Solution

The contribution rates of phenolic compounds were determined by DPPH, ABTS, and FRAP assays. According to the results of phenolic compositions detected by HPLC, polyphenolic model solutions were prepared by the standards according to the detected contents. A phenolic compound was removed from the model solution, and the antioxidant activities of the residual mixture were determined. The formula for calculating the contribution rate of absent substances is ((antioxidant activities value of model solutions − antioxidant activities value of residual solutions)/antioxidant activities value of model solutions) × 100%.

### 3.8. Statistical Analysis

All data were presented as means ± standard deviation (S.D.) with three replications for each prepared sample. The related correlation analyses were expressed by the Pearson’s correlation test. All statistical analyses were performed in SPSS 24.0 for Windows (SPSS Inc., Chicago, IL, USA). *P* < 0.05 was considered statistically significant.

## 4. Conclusions

In the present study, the physicochemical parameters of ZAV including total acids, non-volatile acids, and amino nitrogen of ZAV increased gradually during the brewing process. Reducing sugar decreased sharply at the early fermentation stage and then increased during the AP. Meanwhile, TPC and TFC kept a very low level at the stage of AF and increased to the highest level at the sixth year of aging. TAA showed a similar trend to TPC and TFC, and had a high positive correlation between TPC and TFC during the brewing process. In addition, the contents of *p*-hydroxybenzoic acid, vanillic acid, and catechin increased gradually and reached the highest level at the sixth year of aging, which were the main compositions of phenolic compounds during the AP. Furthermore, we found that gallic acid, ferulic acid, and sinapic acid had a higher contribution at the early stage of the AP, and then declined to a lower level at the latter AP stage. Catechin, vanillic acid, and syringic acid exhibited a gradual increase and then a slight decrease. The contribution rate of each phenolic compound to antioxidant activity may be related to its antioxidant capacity, content, and interactive effect with other compounds. More studies are needed to elucidate the effect of each phenolic compound on TAA and its correlation with other phenolic compounds.

## Figures and Tables

**Figure 1 molecules-24-03935-f001:**
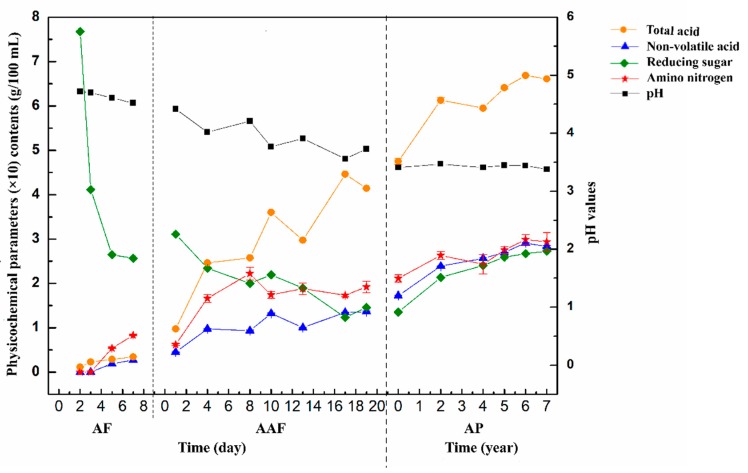
Changes in physicochemical parameters during the brewing process of Zhenjiang aromatic vinegar (ZAV). AF: alcohol fermentation, AAF: acetic acid fermentation, AP: aging process.

**Figure 2 molecules-24-03935-f002:**
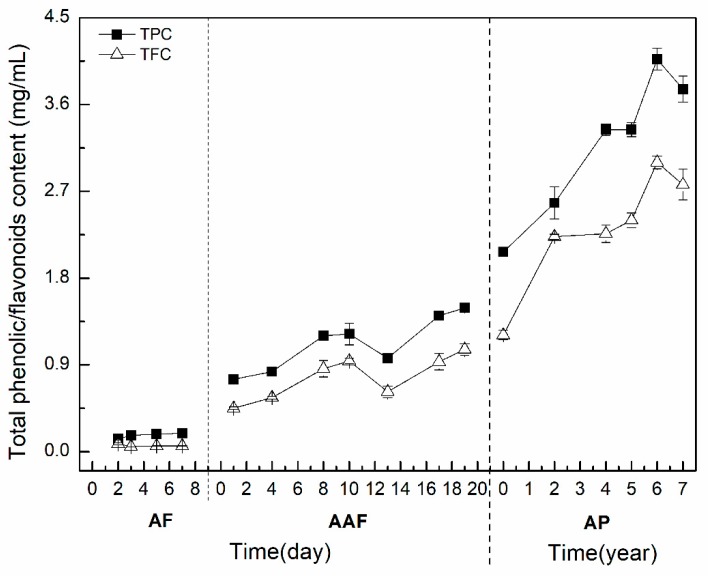
Alteration of TPC and TFC during the brewing process of ZAV. AF: alcohol fermentation, AAF: acetic acid fermentation, AP: aging process, TPC: total phenolic content, TFC: total flavonoid content.

**Figure 3 molecules-24-03935-f003:**
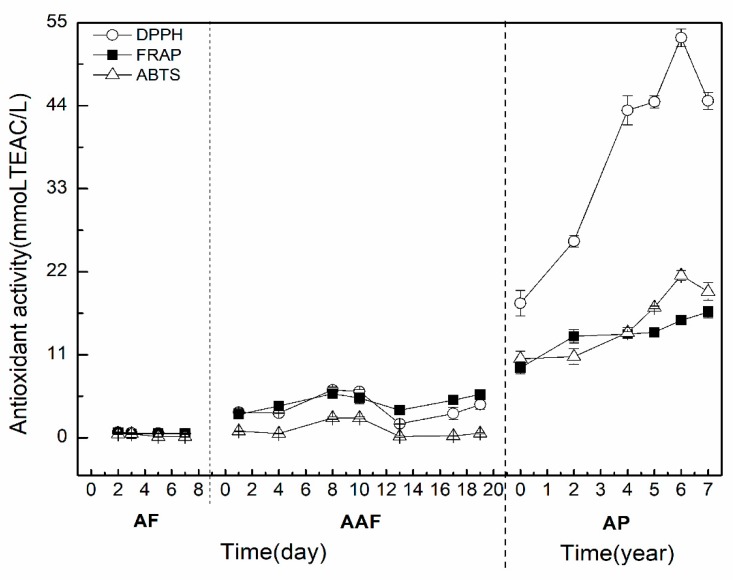
Total antioxidant activity (TAA) and its correlation with TPC and TFC during the brewing process of ZAV. AF: alcohol fermentation, AAF: acetic acid fermentation, AP: aging process.

**Figure 4 molecules-24-03935-f004:**
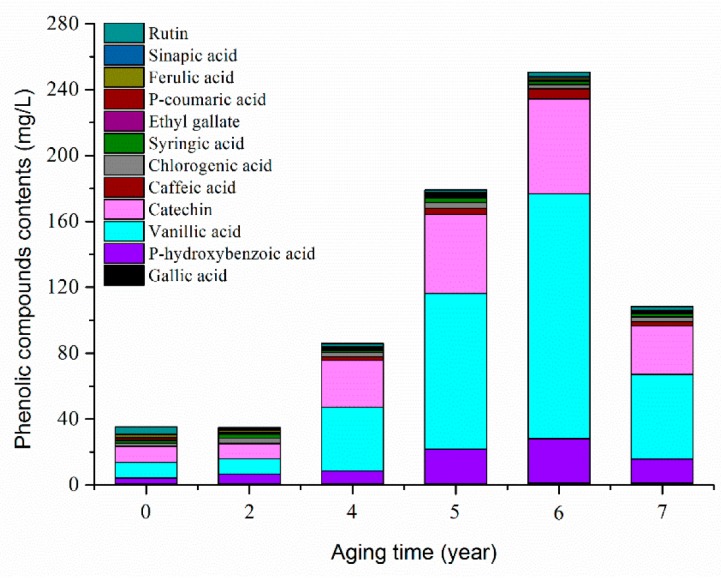
Variation of phenolic compound contents during the AP of ZAV.

**Figure 5 molecules-24-03935-f005:**
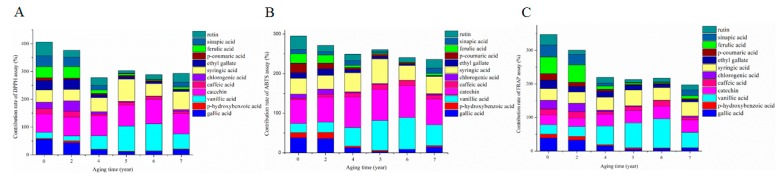
Antioxidant contribution of main phenolic compounds during the aging process of ZAV. Antioxidant activities of phenolic compounds mixture were detected by DPPH assay (**A**), ABTS assay (**B**), and FRAP assay (**C**) during the AP of ZAV.

**Table 1 molecules-24-03935-t001:** Correlation matrix of TPC, TPC, and TAA in ZAV during the brewing process ^1,2^.

	Correlation Matrix				
	TPC	TFC	DPPH Assay	FRAP Assay	ABTS Assay
TPC	1	0.986 **	0.954 **	0.987 **	0.952 **
TFC		1	0.951 **	0.989 **	0.951 **
DPPH assay			1	0.935 **	0.979 **
FRAP assay				1	0.944 **
ABTS assay					1

^1^ Correlations are evaluated using Pearson’s correlation test. ** indicate the significant levels at 0.05, respectively. ^2^ Different numbers represent Pearson correlation coefficient (−1 ≤ r ≤ 1). TPC: total phenolic content, TFC: total flavonoid content, FRAP: ferric reducing antioxidant power.

**Table 2 molecules-24-03935-t002:** The samples of ZAV during the brewing process ^1^.

Alcohol Fermentation (AF) Samples	Acetic Acid Fermentation (AAF) Samples	Aging Process (AP) Samples
2 days	1 day	0 year
3 days	4 days	2 years
5 days	8 days	4 years
7 days	10 days	5 years
	13 days	6 years
	17 days	7 years
	19 days	

^1^ All the origins are in China, Jiangsu.

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
