# Peer review of "Changes of Physicochemical, Bioactive Compounds and Antioxidant Capacity during the Brewing Process of Zhenjiang Aromatic Vinegar"

_molecules, 2019, doi:10.3390/molecules24213935_

Round 1

Reviewer 1 Report

Dear Authors

I am sending you some recommendations, please show it inside the adhesive notes into the paper 

Best regards

Author Response

Response to Reviewer 1 Comments

I am sending you some recommendations, please show it inside the adhesive notes into the paper.

Point 1: Figure 1: please define the abbreviations you use inside the figure, as part of the figure caption.

Response 1: Thanks for your suggestion. As suggested, we have defined the abbreviations in Figure 1. The contents were added as follows:

AF: alcohol fermentation, AAF: acetic acid fermentation, AP: aging process.

The related contents were added in Figure 1 (Page 3 Line 111) of our revised manuscript.

Point 2:  Figure 2: please define the abbreviations (TCP, TFC, AF, AAF, AP) you use inside the figure, as part of the figure caption

Response 2: Thanks for your kind reminding. As suggested, we have defined the abbreviations in Figure 2. The contents were added as follows:

AF: alcohol fermentation, AAF: acetic acid fermentation, AP: aging process, TPC: total phenolic content, TFC: total flavonoid content.

The related contents were added in Figure 2 (Page 4 Line 135-136) of our revised manuscript.

Point 3:  Figure 3: please define the abbreviations (AF, AAF, AP) and acronyms you use inside the figure, as part of the figure caption.

Response 3: Thanks for your kindly suggestion. As suggested, we have defined the abbreviations in Figure 3. The contents were added as follows:

AF: alcohol fermentation, AAF: acetic acid fermentation, AP: aging process.

The related contents were added in Figure 3 (Page 5 Line 157) of our revised manuscript.

Point 4: Table 2: please define the abbreviations (TCP, TFC, AF, AAF, AP) and acronyms you use in Table 2, as part of the captions.

Response 4: Thanks for your kindly reminding. As suggested, we have defined the abbreviations in Table 2. The contents were added as follows:

TPC: total phenolic content, TFC: total flavonoid content.

The related contents were added in Table 2 (Page 5 Line 171-172) of our revised manuscript.

Point 5: Figure 5: please increase the image quality.

Response 5: Thanks for your kindly suggestion. As suggested, we have improved the quality of Figure 5. The related images were shown in Figure 5 as follows:

Figure 5 was replaced in our revised manuscript (Page 7 Line 210). Thanks again for your suggestion to make our manuscript more clearly.

Point 6: Table 1: please, place the corresponding abbreviations also.

Response 6: Thanks for your kindly suggestion. The related contents were shown in Table 1 as follows:

AF: alcohol fermentation, AAF: acetic acid fermentation, AP: aging process.

The related contents were added in Table 1 (Page 7 Line 223) of our revised manuscript.  

Table 1. The samples of ZAV during the brewing process1.

Alcohol fermentation

 (AF) samples

Acetic acid fermentation

 (AAF) samples

Aging process

(AP) samples

2 days

1 day

0 year

3 days

4 days

2 years

5 days

8 days

4 years

7 days

10 days

5 years

13 days

6 years

17 days

7 years

19 days

1 All the origins are in China, Jiangsu.

Point 7: please define the acronym TPTZ in line 235.

Response 7: Thanks for your kindly suggestion. As suggested, “TPTZ” has been corrected as “tripyridyltriazine (TPTZ)” in our revised manuscript (Page 8 Line 256).

Point 8: Abbreviations: please, include the abbreviation AP

Response 8: Thanks for your kindly suggestion. As suggested, we have added the aging process (AP) to abbreviations in our revised manuscript (Page 10 Line 319).

Reviewer 2 Report

The manuscript deals with the antioxidant activity of Zhenjiang aromatic vinegar and the content of phenolic compounds. The topic is very interesting for readers. The manuscript is well written, however some points should be improved:

1.The title and lane 47:"physicochemical and bioactive compounds" - what does it mean? There is something wrong in the title of the manuscript.

2. Introduction: some information about the distribution and consumption of that vinegar worldwide could be presented to readers.

3. Lines : 199-122 and Figure 2: How could  the increase of total phenolic and total flavonoid content during the aging process of vinegar be explained? The reason of the reduction of water content is not convincing because the content of compounds increased approximately by 2-2.5-fold.

Author Response

Response to Reviewer 2 Comments

The manuscript deals with the antioxidant activity of Zhenjiang aromatic vinegar and the content of phenolic compounds. The topic is very interesting for readers. The manuscript is well written, however some points should be improved.

Point 1: The title and line 47:"physicochemical and bioactive compounds" - what does it mean? There is something wrong in the title of the manuscript.

Response 1: Thanks for your kindly reminding. There are many physicochemical and bioactive compounds in vinegar, which are the important parameters to monitor and control vinegar quality during brewing process (Gao, et al; Preventive Nutrition and Food Science 2017 March; 22(1):30-36; Chen, et al; International Journal of Food Properties 2015 August; 19(6):1183-1193). The aim of the study was to investigate the changes of physicochemical properties (total acids, pH, non-volatile acids, reducing sugar and amino nitrogen) and bioactive compounds (total phenolic content (TPC), total flavonoid content (TFC) and phenolic compounds) during the brewing process of Zhenjiang aromatic vinegar (ZAV). As suggested, the title was changed to "Changes of Physicochemical, Bioactive Compounds and Antioxidant Capacity during the Brewing Process of Zhenjiang Aromatic Vinegar ".

Thanks again to make our title more suitable. The related contents were revised in the Title (Page 1 Line 2).

Point 2: Introduction: some information about the distribution and consumption of that vinegar worldwide could be presented to readers.

Response 2: Thanks for your information and suggestion. Vinegar is widely used as an acidic condiment all over the world, which is a kind of fermented and functional food (Solieri, & Giudici; Italy: Springer 2009; pp.1-16). Vinegar can be divided into fruit vinegar and grain vinegar according to different raw materials (Sáizabajo, et al; J. Near Infrared Spectrosc. 2004 January; 12(1):207-219). Most of fruit vinegar is produced by liquid-state fermentation, and mainly distribute in Europe and Africa. These vinegars are made of grapes, apples, tomatoes, persimmons and pineapple and famous as balsamic vinegar, Sherry vinegar and apple vinegar (Daglia, et al; J. Food Compos. Anal. 2013 August; 31(1):67-74; Guerreiro, et al; Anal. Chim. Acta 2014 August; 838:86-92; Madrera, et al; Food Res. Int. 2010 January; 43(1):70-78) Most of grain vinegar is mainly brewed by solid-state fermentation in Asia such as Kurozu vinegar, Shanxi aged vinegar, Zhenjiang aromatic vinegar and Baoning vinegar. The raw materials of these vinegars include sorghum, wheat bran, beans, rice and rice hulls (Nishidai, et al; Biosci., Biotechnol., Biochem. 2000 October; 64(9):1909-1914; Zou, et al; Procedia Chem. 2012 December; 6:20-26).

As suggested, the distribution and consumption of vinegars worldwide were added in Introduction (Page 1 Line 34-40). Thanks again to make our manuscript more complete.

Point 3: Lines: 119-122 and Figure 2: How could the increase of total phenolic and total flavonoid content during the aging process of vinegar be explained? The reason of the reduction of water content is not convincing because the content of compounds increased approximately by 2-2.5-fold.

Response 3: Thanks for your kindly suggestion. In this study, The TPC and TFC were obviously increased with the aging time, which reached the highest at 6-year-old aging time and declined slightly at the 7-year-old aging time. It has been reported that the increased phenolic contents in traditional balsamic vinegar and sherry vinegar are due to the extraction of some phenolic compounds from the wood and evaporation of the water during aging process (AP) (Verzelloni, et al; Food Chemistry 2007 December; 105(2):564-571; Tesfaye, et al; J. Agric. Food Chem. 2002 November 20; 50(24):7053-7061). In addition, flavonol glycosides were hydrolyzed into flavonol aglycones during AP, which can increase the flavonoid content with the aging time (Fang, et al; Food Chem. 2007 December; 101(1):428-433). Briefly, the increase of TPC and TFC is likely associated with the reduction of water content and hydrolyzation of flavonol glycosides during the AP.

The related contents were revised in Results and Discussion (Page 3 Line 125-126; Page 4 Line 127-132) of our revised manuscript.

Reviewer 3 Report

I appreciate the authors to have made an effort to conduct a nice study centering around physicochemical changes during specialized vinegar fermentation and ageing. However, the study, while could be useful to peers in the field, lacks significant scientific value and requires more information to be sufficiently translatable within international communities. Based on comments discussed below, I recommend major revision of the manuscript and would significantly recommend the authors to consider the comments constructively and improve their work to be submitted for publication in the journal. My specific comments are as follows:

The language used in the manuscript is very bad and needs extensive editing. My suggestion is that the authors require help from a language expert to fix the manuscript. Almost every sentence in the early parts of the manuscript has grammatical errors in it. 

Line 38: "alcoholic fermentation"?? I thought that we were talking about vinegar - The authors should first indicate what are all the steps involved in the vinegar production.

What is pei? It does not seem to be an internationally used term during fermentation and to be appealing for an international community, the authors should describe what they mean when they use this term.

The authors indicate that the phenolic activities of grape and apple vinegars are almost 100-1000 times greater than that in rice vinegars. It is still unsure what is the use of doing this study? Are the authors expecting higher phenolic activities comparable to grape and apple vinegars? There is no evidence of it.

It seems that the authors did not do the brewing process but instead got the samples and analyzed the compositions in it. I doubt that this study would be scientifically significant or reproducible since the authors do not give any parameters of the brewing process. At this low pH, how did the total acid keep increasing and is there an indication that all this acid is vinegar and not other acids produced during fermentation? The reason for this question is what was the specific acetic acid concentration in the vinegar during the brewing process at the sample points that showed different phenolic activities? 

Line 139: Precipitation??

Line 166-169: It is understandable that temperature changes are common during season changes. However, without sufficient data, there is no indication that the temperature change is significant enough to result in Maillard browning. Neither do we know that the browning effect was similar every year. Since this is not a controlled process, the data could also be variable between batches set in different years. The study, however, was done with different batches set in the same year. Hence, the study lacks controls confirming that their observed trends is similar for every batch of ZAV every year.

THe authors keep bringing up Maillard browning and meladoinins. There is no data indicating presence of meladoinins except that the authors believe some of the phenolics compounds could result in formation of these compounds. The discussion is not confirmed by the data.

It is still unclear how many replicates were done to measure standard errors. The authors indicate that there were more than one sample per data point but does not indicate the exact amount and whether the amount was statistically significant (i.e., alpha of the samples).

Author Response

Response to Reviewer 3 Comments
I appreciate the authors to have made an effort to conduct a nice study centering around physicochemical changes during specialized vinegar fermentation and ageing. However, the study, while could be useful to peers in the field, lacks significant scientific value and requires more information to be sufficiently translatable within international communities. Based on comments discussed below, I recommend major revision of the manuscript and would significantly recommend the authors to consider the comments constructively and improve their work to be submitted for publication in the journal. My specific comments are as follows:

Point 1:  The language used in the manuscript is very bad and needs extensive editing. My suggestion is that the authors require help from a language expert to fix the manuscript. Almost every sentence in the early parts of the manuscript has grammatical errors in it. 

Response 1: Thanks for your kindly suggestion. As suggested, we have carefully proofed our manuscript and improved our expression. Grammatical and typing errors were revised by a language expert. 

Point 2: Line 38: "alcoholic fermentation"?? I thought that we were talking about vinegar - The authors should first indicate what are all the steps involved in the vinegar production. 

Response 2: Thanks for your kindly suggestion. We are sorry for the unclear description. Production technologies of vinegars are classified as solid-state fermentation and liquid-state fermentation (Xu, et al; Food Microbiol. 2011 September; 28(6):1175-1181; Bertelli, et al; Food Anal. Method. 2015 February 8(2):371-379). Zhenjiang aromatic vinegar (ZAV), one of the famous Chinese vinegar, is made of grains such as sticky rice, barley, wheat bran and rice hulls (Xu, et al; Journal of the Science of Food and Agriculture 2011 July; 91(9):1612-1617). ZAV is produced by solid-state fermentation techniques including saccharification and alcoholic fermentation (AF), acetic acid fermentation (AAF), leaching, decoction and aging (Liu, et al; Journal of Functional Foods 2016 March 2016; 21:75-86). During the AF, the raw materials were crushed, cooked, saccharificated and fermented. 
The related contents were revised in Introduction (Page 1 Line 40-43) of our revised manuscript.

Point 3:  What is pei? It does not seem to be an internationally used term during fermentation and to be appealing for an international community, the authors should describe what they mean when they use this term.

Response 3: Thanks for your information and suggestion. We are sorry for the unclear description. AAF of ZAV is a typical process of multispecies solid-state fermentation. Vinegar Pei, a special term in AFF of Chinese vinegars, refers to a mixture of alcohol mash, raw materials and complex bacterial communities (Peng, et al.; Current Microbiology 2015 August; 71(2):195-203). 
 The related contents were revised in Introduction (Page 2 Line 44-46) of our revised manuscript. Thanks again to make our manuscript more clearly and complete.

Point 4:  The authors indicate that the phenolic activities of grape and apple vinegars are almost 100-1000 times greater than that in rice vinegars. It is still unsure what is the use of doing this study? Are the authors expecting higher phenolic activities comparable to grape and apple vinegars? There is no evidence of it. 

Response 4: Thanks for your kindly reminding. In this study, the unit of antioxidant activities in grape and apple vinegars is not uniform with that in ZAV. Bakir et al. reported that the antioxidant activities of grape vinegar and apple vinegar by 2,2’-azino-bis(3-ethylbenzthiazoline-6-sulfonic acid (ABTS) assay ranged from 418 ± 49 mg TEAC/100 mL to 2561 ± 260 mg TEAC/100 mL, which were equal to 16.6 mmol TEAC/L-102.3 mmol TEAC/L. In our study, the antioxidant activities of ZAV during the aging process (AP) were 11.85-21.25 mmol TEAC/L by ABTS assay. The antioxidant activities in grape and apple vinegars are 1-5 times higher than that in ZAV. In addition, ZAV samples in the present study were obtained from industrial ZAV during the AP. In our previous study (Zhao, et al; Molecules 2018 November; 23(11):2949), the antioxidant activities of ZAV from traditional ZAV during the AP were from 11.96 ± 0.77 to 41.07 ± 1.06 mmol TEAC/L, which were higher than that from industrial ZAV. This study focused on the kinds and contents of phenolic compounds and TAA in vinegars. The aim of this study was to investigate the changes of total phenolic content (TPC), total flavonoid content (TFC) and total antioxidant activity (TAA) were determined during the brewing process of ZAV, and explore the correlation between phenolic compounds contents and antioxidant activities during the AP of ZAV. 

Point 5: It seems that the authors did not do the brewing process but instead got the samples and analyzed the compositions in it. I doubt that this study would be scientifically significant or reproducible since the authors do not give any parameters of the brewing process. At this low pH, how did the total acid keep increasing and is there an indication that all this acid is vinegar and not other acids produced during fermentation? The reason for this question is what was the specific acetic acid concentration in the vinegar during the brewing process at the sample points that showed different phenolic activities?

Response 5: Thanks for your questions. In the present study, we did not do the brewing process of ZAV. ZAV samples during the brewing process were obtained from Jiangsu Hengshun Vinegar Industry Co., Ltd. (Jiangsu, China). Physicochemical parameters of the brewing process including total acids, non-volatile acids, reducing sugar, amino nitrogen and pH were detected and showed in Figure 1. 
Vinegar contains a specified amount of organic acids that include acetic acid, tartaric acid, formic acid, lactic acid, citric acid, malic acid, and succinic acid (Cocchi, et al; Talanta 2006 August; 69(5):1166-1175). Among them, acetic acid is the main ingredient of vinegar, account for about 30 %-50 % in total organic acids (Chen, et al; Acetic Acid Bacteria 2013 January; 2(s1):e6; Wang, et al; Ultrason Sonochem 2017 November; 39:272-280). In this study, total acids were measured by acid-base titration according to GB/T 12456-2008, and acetic acid was used as a reference compound to expressed as total acids of vinegar (Chen, et al; J. Food Sci. 2012 February; 77(2):C222-227). Total acids refer to the sum of the concentration of ionized and unionized hydrogen. pH of ZAV samples was determined by a pH meter (Metrohm, Herisau, Switzerland). pH is related to the concentration of ionized hydrogen, which can be calculated through formula: pH= -log10 {H+} (Magrí, et al; Environmental Technol. 2007 April; 28(3):255-265). Therefore, the total acids of vinegar are not only related to the concentration of ionized hydrogen, but correlated with unionized hydrogen. The trend of total acids is not certainly consistent with the trend of pH in ZAV samples during the brewing process.
Moreover, acetic acid concentration is not correlated with phenolic activities in vinegar. Many studies have reported that the antioxidant capacity of vinegar has a highly positive correlation with polyphenols and flavonoids contents (Xie, et al; J. Food Sci. 2017 October; 82(10):2479-2486; Verzelloni, et al; Food Chem. 2007 December; 105(2):564-571). Therefore, TAA and its correlated with TPC, TFC were evaluated in the present study.

Point 6:  Line 139: Precipitation??

Response 6: We are sorry for the unclear description. During the AP, the low pH value and the decrease of the water content may promote the formation of the oligomers and polymers, which subsequently evolve to more polymerized compounds and finally precipitate (Verzelloni, et al; J. Food Biochem. 2010 May; 34(5):1061-1078). Several studies have reported that phenolic compounds can interact with macromolecules such as melanin, proteins and polysaccharides to form precipitates in vinegars during the AP (Bourvellec, & Renard; Crit. Rev. Food Sci. Nutr. 2012 March; 52(3):213-248; Bindon, et al; Food Chem. 2016 December; 199:838-846). Therefore, the decrease of TPC, TFC is likely due to the increased precipitation during the AP, which subsequently declined TAA at this stage.
The related contents were revised in Results and Discussion (Page 4 Line 149-153) of our revised manuscript. 

Point 7: Line 166-169: It is understandable that temperature changes are common during season changes. However, without sufficient data, there is no indication that the temperature change is significant enough to result in Maillard browning. Neither do we know that the browning effect was similar every year. Since this is not a controlled process, the data could also be variable between batches set in different years. The study, however, was done with different batches set in the same year. Hence, the study lacks controls confirming that their observed trends is similar for every batch of ZAV every year.

Response 7: Thanks for your information and suggestion. The Maillard reaction occurring between an amino acid or protein and a reducing sugar is a ubiquitous food reaction that takes place during storage, cooking and heat processing (Martins, et al; Trends Food Sci. Technol. 2001 January; 11(9-10):364-373). Melanoidins, dark-brown polymers, are major compounds generated in the late stage of the Maillard reaction (Moreira, et al; Food Funct. 2012 April; 3(9):903-915; Wang, et al; Food Chem. 2011 March; 128(3):573-584). Melanoidins in vinegar are mainly produced during thermal and aging processes (Wang, et al; Flavour Fragrance J. 2012 January; 27(1):47-53). During the AP, the low pH value and the decrease of the water content by evaporation may promote the formation of polymeric compounds including melanoidins (Verzelloni, et al; J. Food Biochem. 2010 May; 34(5):1061-1078). In addition, in the present study, ZAV samples were obtained from industrial processing line at different time points from different batches set in the same year. The conditions in the industrial processing line are better controlled than those in the traditional processing line. More studies will be carried from every batch of ZAV in every year in future. 

Point 8: The authors keep bringing up Maillard browning and meladoinins. There is no data indicating presence of meladoinins except that the authors believe some of the phenolics compounds could result in formation of these compounds. The discussion is not confirmed by the data. 

Response 8: Thanks for your kindly suggestion. As suggested, we have been added data indicating presence of meladoinins in our revised manuscript. The contents were added as follows:
2. Result and Discussion
As shown in Table S1, the browning index (A420 nm) of high molecular weight melanoidins ranged from 0.26 ± 0.01 to 0.83 ± 0.03, which reached highest at 6th year of aging and decreased slightly at 7th year of aging. The increase of the browning index indicates that the Maillard reaction continues and the synthesis of polymeric compounds increases during AP of 6 years.
Browning index was also added in Table S1 as follows:
Table S1. Browning index of ZAV samples during the aging process1, 2.
Aging process (AP) samples Browning index (OD 420 nm) 
0 year 0.26 ± 0.01 a
2 years 0.53 ± 0.01 b
4 years 0.67 ± 0.01 c
5 years 0.71 ± 0.03 d
6 years 0.83 ± 0.03 f
7 years 0.74 ± 0.06 e
1 Data are presented as mean ± S.D. (n = 3). 2 Significant differences are evaluated using the Duncan Multiple comparison Test. Different letters in the column presents statistically significant differences (p < 0.05). 
3. Materials and Methods
Determination of high molecular weight melanoidins
Vinegar samples were diluted and filtered with filter papers 40 (Whatman, Maidstone, UK). Each filtered sample was ultra-filtrated with Amicon Ultra-4 regenerated cellulose 10 kDa (Millipore, MA, USA) and centrifuged at 7500 g for 20 min at 4 ℃. High molecular weight melanoidins were isolated from each filtered sample. The amount of high molecular weight melanoidins was determined as browning index. The absorbance was detected at 420 nm. 
The related contents were revised in Results and Discussion (Page 6 Line 182-184 and Line 188-189), Materials and Methods (Page 9 Line 275-280) and Supplement Materials (Page 9 Line 309-310) of our revised manuscript.

Point 9: It is still unclear how many replicates were done to measure standard errors. The authors indicate that there were more than one sample per data point but does not indicate the exact amount and whether the amount was statistically significant (i.e., alpha of the samples).

Response 9: Thanks for your kindly suggestion. Thanks for your kindly suggestion. We are sorry for the unclear description. There were four samples per time point, and each sample with three replicates to measure standard errors. All data were presented as means ± standard deviation (S.D.) with three replications for each prepared sample. P < 0.05 was considered statistically significant.
The related contents were added in Materials and Methods of our revised manuscript (Page 9 Line 290-291 and Line 292-293). Thanks again for your suggestion to make our manuscript more clearly.

Round 2

Reviewer 2 Report

The manuscript has been improved and now is suitable for publication.

Reviewer 3 Report

The authors have made sufficient changes to the manuscript especially the addition of comparisons between ZAV antioxidant assay and that for grape and apple vinegars was not even within the same scale and very confusing. Also, the addition of data from browning index helps there conclusions on meladonins and Maillard browning process. The manuscript seems to be well set and I am happy to recommend acceptance for publication.